# Sand mining far outpaces natural supply in a large alluvial river

Christopher R. Hackney[1], Grigorios Vasilopoulos[2], Sokchhay Heng[3], Vasudha Darbari[2], Samuel Walker[2], Daniel R. Parsons[2]

[1] School of Geography, Politics and Sociology, Newcastle University, Newcastle upon Tyne, NE1 7RU, UK
[2] Energy and Environment Institute, University of Hull, Hull, UK, HU6 7RX
[3] Institute of Technology of Cambodia, Phnom Penh, Cambodia

*Correspondence to*: Christopher R. Hackney (christopher.hackney@ncl.ac.uk)

**Abstract** The world's large rivers are facing reduced sediment loads due to anthropogenic activities such as hydropower development and sediment extraction. Globally, estimates of sand extraction from large river systems are lacking, in part due to the pervasive and distributed nature of extraction processes. For the Mekong River, the widely assumed estimate of basin-wide sand extraction is 50 Mt per year. This figure is based on 2013 estimates and is likely to be outdated. Here, we demonstrate the ability of high-resolution satellite imagery to map, monitor and estimate volumes of sand extraction on the Lower Mekong River in Cambodia. We use monthly composite images from PlanetScope imagery (5 m resolution) to estimate sand extraction volumes over the period 2016 – 2020 through tracking sand barges. We show that rates of extraction have increased on a yearly basis from 24 Mt (17 Mt to 32 Mt) in 2016, to 59 Mt (41 Mt to 75 Mt) in 2020 at a rate of ~8 Mt yr$^{-1}$ (6 Mt yr$^{-1}$ to 10 Mt yr$^{-1}$), where values in parenthesis relate to lower and upper error bounds, respectively. Our revised estimates for 2020 (59 Mt) are nearly two times greater than previous best estimates for sand extraction for Cambodia (32 Mt) and greater than current best estimates for the entire Mekong Basin (50 Mt). We show that over the five year period, only two months have seen positive (supply exceeds extraction) sand budgets under mean scenarios (five months under the scenarios with greatest natural sand supply). We demonstrate that this net negative sand budget is driving major reach-wide bed incision with a median rate of -0.26 m a$^{-1}$ over the period 2013 to 2019. The use of satellite imagery to monitor sand mining activities provides a low-cost means to generate up-to-date, robust estimates of sand extraction in the world's large rivers that are needed to underpin sustainable management plans of the global sand commons.

## 1 Introduction

The world's rivers transport ~37,000 km$^3$ of water (Dai and Trenberth, 2002) and ~19 billion tonnes of sediment and minerals (Milliman and Farnsworth, 2011) to the global oceans each year. Over the past century, anthropogenic pressures associated with population growth, urbanisation and economic development have seen an increase in the demand for the natural resources river systems provide (Best, 2019; Best and Darby, 2020). In particular, the demand for sand and gravel from rivers has seen unprecedented growth since the 1970s (Peduzzi, 2014; Miatto et al., 2017; Torres et al., 2017), with demand now outstripping

supply (Bendixen et al., 2019). Estimates indicate that today, at a minimum, between 32 and 50 billion tonnes of aggregates are extracted globally every year (Koehnken and Rintoul, 2018). The environmental impacts of such high extraction rates are being felt around the world (Koehnken et al., 2020), with impacts manifest as river bank instability (Hackney et al., 2020), changes in riverine biodiversity (Bhattacharya et al., 2019), increased tidal incursion (Vasilopoulos et al., 2021) and saline

intrusion in deltaic environments (Eslami et al., 2019), amongst others (Koehneken et al., 2020; Bisht, 2021). Yet, our inability to monitor levels of extraction, or the locations in which extraction occurs, routinely and accurately, currently hinders our ability to assess the sustainability of global sand extraction and provide effective environmental impact assessments for regions undergoing extensive mining.

In order to effectively and sustainably manage riverine sand resources, accurate and regular monitoring of the locations and rates of sediment extraction are required (Peduzzi, 2014; Bendixen et al., 2019). Yet the pervasive and diffuse nature of the riverine extraction process means that it often occurs at spatial scales (spread across many $km^2$ of river surface area), and in remote locations, that make on-site monitoring difficult. The advent of regular, high-resolution satellite imagery technologies (now often up to 3m ground pixel resolution and at sub-weekly return periods) permits the remote monitoring of the locations

and extent of sediment mining in riverine and lacustrine environments (Duan et al., 2019; Xin and Park, 2021). As such, it is now possible to provide a synoptic overview of the locations and rates of sediment mining across large (basin and sub-catchment scale) areas.

The Lower Mekong River (LMR) is currently experiencing levels of sand mining that exceed natural sediment supply by up

to a factor of seven (Hackney et al., 2020). This is resulting in net reach-wide incision (Hackney et al., 2020; Jordan et al., 2019, Vasilopoulos et al., 2021) with associated impacts on tidal extent (Vasilopoulos et al., 2021) and saline intrusion rates (Eslami et al., 2019; Loc et al., 2021), changing flood frequencies (Park et al., 2020), impacts on regional fish stocks (Nuon et al., 2020) and alterations to the hydrology of the Tonle Sap Lake (Xin and Park, 2021). The current existing estimates of rates of sand extraction for the Mekong basin are placed at around 50 Mt yr$^{-1}$ (31 million m$^3$ yr$^{-1}$ assuming a density of sand of 1,600

kg m$^{-3}$ following Bravard et al., 2013). This estimate is based on questionnaire data at mining sites carried out in 2011 and 2012 (Bravard et al., 2013). Since then, however, the demand for aggregates is likely to have increased, and so these estimates may be considerable underestimates of current extraction rates. Recent work has quantified one component of the LMRs sand budget, that of the incoming sediment load (Hackney et al., 2020). However, tighter constraints on the locations and rates of extraction, and exactly how these have changed since the estimates of Bravard et al. (2013), are lacking.


Here, we detail the application of high-resolution (~5 m) PlanetScope satellite imagery to map and monitor the locations and extraction activity throughout the Lower Mekong River (LMR; Figure 1) in Cambodia over the period 2016 – 2020. We detail how this methodology permits, for the first time, the identification of extraction hotspots, and we provide revised estimates of the volumes of sand extracted through the LMR in Cambodia. We combine these estimates of extracted volume with estimates

of sand transport through the LMR and demonstrate the level of sand deficit (the volume of sand naturally transported by the LMR minus volume extracted) being experienced throughout the LMR. Furthermore, we combined satellite image analysis and targeted bathymetric surveys to highlight the morphological impact of ongoing sand mining in the LMR.

## 2 Study site and Methods

### 2.1 The Mekong River

The Mekong River is one of the most intensely studied large river reaches in the world due to its combination of development needs and human interference with its natural system (Bussi et al., 2021; Darby et al., 2016; Hackney et al., 2020; Räsänen et al., 2017; Schmitt et al., 2017). Draining the Tibetan Plateau, the Mekong delivers 450 km$^3$ yr$^{-1}$ of water (MRC, 2009) and 87 Mt yr$^{-1}$ of sediment (Darby et al., 2016) to the South China Sea (Figure 1a). Of that incoming suspended sediment load only 6 Mt yr$^{-1}$ is sand (Hackney et al., 2020) due in part to extensive sediment trapping upstream in the Mekong catchment, with

hydropower development retaining up to 96% of the catchment sediment load (Kondolf et al., 2014, 2018; Schmitt et al., 2017). This trapping effect is compounded by climatic changes altering the monsoon-driven hydrology of the LMR, resulting in reduced suspended sediment loads  (Darby et al., 2016; Walling and Fang, 2003) and driving changes to the morphology and flows of the river (Anthony et al., 2015; Brunier et al., 2014; Eslami et al., 2019; Ha et al., 2018; Vasilopoulos et al., 2021). Prior estimates based on the study by Bravard et al. (2013) suggest that annually 35 million cubic meters (56 Mt) of sand,

gravel and pebbles are removed from the Mekong river in its entirety every year. Of this volume, sand accounts for 90% (31 million cubic meters, or 50 Mt, per year). At the time, Cambodia accounted for 60% of the Mekong's sand extraction (18.7 million cubic meters or 32 Mt per year; Figure 1a) with the 18.1 million cubic meters (29 Mt) being extracted between Kampong Cham and the Vietnamese border (Figure 1b; Bravard et al., 2013). In the years since this initial estimate, Phnom Penh (along with many other cities in Cambodia and within the LMR basin) has seen rapid urban development (Mialhe et al.,

2019) with widespread infilling of floodplain lakes and land reclamation projects, all of which are expected to have increased local demand for sand. As such, there is a vital need for more up to date estimates of sand extraction for the Mekong to account for changes over the past decade.

### 2.2 Identification of mining vessels from satellite imagery

Monthly composite images from Planet labs PlanetScope Surface Reflectance (Planet Team, 2018) product were used to

identify boat activity on the Mekong River in Cambodia south of Kampong Cham to the Vietnamese border for the period January 2016 to December 2020 (Figure 1). These monthly composites provide high resolution (4.77 m at the equator) red (590 – 670 nm) green (500 – 590 nm) blue (455 – 515) images with minimal cloud cover projected in WGS84 Mercator (Figure 2a).  This product was chosen to ensure that cloud cover effects were minimised throughout the monsoon periods (June – October) and thus avoid seasonal bias in our extraction estimates (see below). The resolution of the composite images (4.77

m) is suitable for identifying sand mining vessels as these are typically ~60 m in length and ~10 m wide, thereby covering

multiple pixels within the composite images (Figure 2). Smaller vessels such as fishing vessels and ferries were too small to be consistently identified within the composite scene, whilst larger shipping and container vessels are clearly visible yet distinguishable due to their larger size and location. The main trading port is located south of Phnom Penh and this container traffic is limited to the areas downstream of the port. As such, we can say with confidence that the majority of water borne traffic identifiable from the composite images within the area of interest is related to mining activity (Figures 1 and 2).

## 2.3 Estimation of extraction volumes

Within each monthly composite the centre of each vessel visible is identified based on their colour and shape (Figure 2b) and recorded in QGIS. For each monthly time step, the total number of boats visible is counted. All vessels within a 100 m buffer from the riverbanks are removed from the monthly count as it is assumed these boats are moored and are deemed inactive. As the vessels may have a length up to ~75 m (Figure 2), and visual observations suggest that active mining operations do not occur along the near bank zone, the 100 m buffer was chosen to ensure that if moored perpendicular to the bank, vessels would be fully contained within, whilst ensuring active operations remained outside, the buffer. Along the study reach, the channel width varies between 600 m at its narrowest to 2,500 m at its widest – although typically channel widths are around 900 m. A 100 m buffer on each bank, therefore accounts for between 8 and 33% (though typically 22%) of the channel width ensuring the majority of the channel, and the areas that are actively mined, remain accounted for in our subsequent analysis. Of those boats deemed active (i.e. more than 100 m away from the riverbank), it is assumed that each vessel is filled and emptied once a day. This assumption is based on visual observations of the time it takes for a vessel to be completely filled by a dredging platform (approximately 3 hours) meaning it is unlikely that the same boat is filled more than once a day. The price of sand currently limits supply chains to local sources, as large distance supply chains are not economically viable. Therefore, we assume that each filled boat transits only a relatively short distance (i.e. to Phnom Penh, and a maximum distance of 50 km) before unloading, allowing for the filling and unloading cycle to be completed in one day.

Using high resolution satellite imagery available in Google Earth (provided by CNES/Airbus on 24/09/2019, which covered the entire study area), the dimensions of sand mining vessels were constrained. Across 50 vessels, the vessel length (m), $L_b$, vessel width (m), $W_b$, the length of the vessel hold (m), $L_h$, and the width of the vessel hold (m), $W_h$, were recorded (Figure 2c). Google Earth imagery was used to constrain the vessel dimension as its higher spatial-resolution permits greater confidence in providing robust dimensions compared to the PlanetScope imagery. Field observations of vessel height ($H_v$) were taken of vessels in dry dock for maintenance. We estimate the volume each mining vessel can hold as 2100 m$^3$ ($L_h$ x $W_h$ x $H_v$; 29.3 m x 8.1 m x 7 m). A sand density of 1,600 kg m$^{-3}$ is used to convert this volume into a tonnage applying a compaction factor of 0.5 to account for the space left between grains due to the unconsolidated nature of the sand being dredged. We use the same density value used by Bravard et al. (2013) to permit direct comparison with their estimates of tonnage extracted. We note here that using the density for quartz sand (2,650 kg m$^{-3}$) would result in a 65% increase in our subsequent estimates of tonnage extracted. In determining sand transport on the river bed from multibeam echo surveys Nittrouer et al. (2008) and

Hackney et al. (2020) use a bed compaction factor ($C_b$) of 0.35. The value used here is higher as the force of the disturbance

of suction dredging sand, its deposition in the mining vessels and the subsequent draining of water from the material is likely

to reduce pore spaces and increase compaction. As there is no other compaction aside from gravity, but less water within the

deposited material, we assume a reduced pore space between sand grains within the vessel compared to the riverbed. To

account for uncertainty we vary $C_b$ with an upper value of 0.65 and the lower limit of 0.35. We subsequently report all values

using a central estimate of $C_b$ of 0.5 with upper and lower bounds in parenthesis. For each month, the daily tonnage values

calculated from the active vessel count are multiplied by the number of days in each calendar month, assuming vessels are

active seven days a week. These are then summed across the year to provide annual estimates of sand extraction.

## 2.4 Estimation of sediment transport rates for 2016 – 2020

Suspended- and bedload- sand transport was estimated for the LMR for the period 2016 to 2020 using ratings curves published

in Darby et al. (2016) and Hackney et al. (2020). These estimates are derived for the monitoring station at Kratie (Figure 1)

which is approximately 250 km upstream from Phnom Penh. Daily water levels ($d$, m) and discharge ($Q$, m³ s⁻¹) for Kratie are

available for the period 2016 – 2020 and 2016 – 2018, respectively, from the Mekong River Data Portal

(portal.mrcmekong.org; station identifier 014901). To extend $Q$ up to 2020, a ratings curve ($r^2 = 0.96$, $p < 0.05$) is derived for

the period of data overlap (2016 – 2018) where:

$$Q = 29.149d^{2.4059} \tag{1}$$

Suspended sediment flux ($Q_s$, kg s⁻¹) was then calculated using the ratings curve for Kratie from Darby et al. (2016), where:

$$Q_s = 0.0001298Q^{1.763} \tag{2}$$

Of this total suspended sediment flux, only 7% is composed of the sand fraction (Hackney et al., 2020). As such, the suspended

sediment fluxes are multiplied by 0.07 to derive a suspended sand flux. Prior fieldwork across a range of discharges (14,500

to 55,000 m³ s⁻¹) and a range of study sites located within the study region (Hackney et al., 2020) shows that the range of sand

fraction within the water column does range from 1 to 14% (averaging out across the study area at 7%). Thus it is likely that

the sand fraction varies spatially and temporally (as discharge varies). We acknowledge that this variation will lead to

substantial changes in the estimates of the sand deficit (natural transport vs. extraction) and that locally, sand deficits may be

greater and lower depending on the availability of sand in suspension.


Bedload transport ($Q_b$) was calculated using the bedload ratings curve of Hackney et al. (2020):

$$Q_b = 0.0049233\omega_0^{1.4252} \tag{3}$$

where $\omega_0$ is the critical stream power calculated as

$$\omega_0 = = 290D_{50}^{\frac{3}{2}} \log\left[12\frac{d}{D_{50}}\right] \tag{4}$$

Where $D_{50}$ is the median grain size (m) reported as being 500 μm, defined by bed sediment samples collected with an Ekman-style grab sampler. Each sample was dry-sieved for grains >75 μm, while finer grains were analysed using a Saturn DigiSizer to estimate the full grain-size distribution of each sample. The development of the empirical bedload function employed is reported in detail in Hackney et al. (2020), but for brevity here, the method is based on that of Bagnold (1980). Statistically, the model has an $r^2$ consistent with that of Bagnold's (1980) model ($r^2 = 0.22$, $P < 0.1$) based on 12 observations across a range of discharges covering 78% of all bedload transport events during the period 1980 - 2015. As such, given the ever difficult nature of characterising bedload sediment transport, we have confidence that the model used here is appropriate for the study reach. Bedload transport has been shown to be a small component of the overall sand transport in the Mekong, contributing $0.18 \pm 0.07$ Mt yr$^{-1}$ (Hackney et al., 2020); as such the greatest uncertainty in estimating the natural transport of sand within the Mekong comes from the estimates of the suspended sand load which is estimated to contribute $6.18 \pm 2.01$ Mt yr$^{-1}$ (Hackney et al., 2020).

**2.5 Bathymetric change detection**

To highlight bathymetric change resulting from sand mining activities, river bed bathymetry from 2013 and 2019 are compared. A set of single beam echo sounding (SBES) surveys of the Mekong River from Neak Loeung to Kratie (Figure 1) was conducted between September and November 2013. A Garmin Fishfinder was connected to a Trimble differential-GPS unit and bathymetry was recorded at a rate of 1 Hz. Survey lines were designed in a saw-tooth pattern, completing a cross-section of the channel at one channel width spacing. Water elevation was calculated based on the water surface slope between the two Mekong River Commission (MRC) gauges and the water level at the closest MRC gauge at the date of the survey. Daily gauge readings for each site were obtained over the survey period. The daily values were then interpolated to hourly values using a cubic spline interpolant. The water surface slope between the two closest gauges for each hour was calculated assuming a linear change in water surface slope between gauges. Distances between each gauge were calculated along the river centreline. For each hour, the water surface slope was applied to the closest gauges' water level reading to generate a variable water level for each survey date. To convert water depths to channel bed elevations, the zero gauge height at Kampong Cham was used referenced to WGS84 ellipsoid. All water depth readings were adjusted to the Kampong Cham zero gauge height, determined from a dGPS survey of the Kampong Cham gauge conducted in November 2013. Depending upon which MRC gauge was closest, different offsets were applied (Kratie = +0.15 m, Neak Loeung = -0.6 m). The elevation data were used to generate a raster DEM using a Kriging interpolant at 5m resolution.

Additionally, a bathymetry survey of the Mekong River at Rokar Korng (see Figure 1) was conducted in June 2019. A Sontek M9 acoustic Doppler current profiler (ADCP) equipped with real-time kinematic GPS (RTK-GPS) was used to collect riverbed elevation (below water surface), whilst differential GPS was used to collect riverbank elevation including islands (above water surface). The bathymetry at each survey point was obtained by subtracting water depth recorded on the M9s 1 MHz vertical beam from water surface elevation measured by the RTK-GPS. The reference datum of the RTK-GPS (WGS84) was used to

register the water surface elevation recorded nearest to the river bank. The point elevation dataset was kriged to a 5m raster for comparison with the 2013 dataset.

## 3 Results

### 3.1 Temporal variations in sand extraction (2016 – 2020)

Over the period of 2016 – 2020 the number of sand mining vessels operational and active (i.e. more than 100 m from the river bank) on the Mekong varied from twelve in May 2016 to 143 in November 2020 (Figure 3). An increase in the number of vessels in operation is evident over the five year period, with superimposed seasonal changes in boat activity mapping onto periods of high flow. During periods of high water levels (blue line in Figure 3) corresponding drops in boat numbers are observed, for example during the periods of August 2018 and August 2019, as conditions become unfavourable for boat travel on the river during the high flow monsoon periods.

Converting the number of active vessels observed each month to a volume of sand (see section 2.3) and integrating these estimates over the calendar year reveals a year on year increase in the volume of extracted sand from the Mekong River between Kampong Cham and the Vietnamese border (Figure 4). Estimates of sand extracted for 2016 reveal that 24 Mt (17 Mt to 32 Mt) of sand was removed, with numbers in parenthesis representing upper and lower bounds (see section 2.3). Our 2016 estimate is consistent with the estimate of Bravard et al. (2013) who placed the extraction levels in Cambodia for 2013 at 30 Mt yr$^{-1}$. However, in the subsequent years, the volume of sand extracted from this reach of the Mekong has risen significantly to an estimated 59 Mt (41 Mt to 75 Mt) in 2020 (Figure 4), at an increase of ~ 8 Mt yr$^{-1}$ (6 Mt yr$^{-1}$ to 10 Mt yr$^{-1}$). The 2020 extraction volume is nearly double that reported by Bravard et al. (2013) for Cambodia and is comparable to the extraction rates estimated for the entire Mekong basin in 2013 (50 Mt yr$^{-1}$).

Over the same period, however, the natural supply of sediment within the Mekong River has not kept pace with extraction rates (Figure 4 and 5). The greatest annual volume of sand transported by the river in the period of observation was in 2018, when it reached an estimated 10.9 Mt (0.8 Mt to 30 Mt). Comparing an extraction volume of 50 Mt (35 Mt to 65 Mt) for 2018 to the natural supply, nearly five (two to fourty three) times the volume of sand supply is being removed. Moreover, natural sand supply is observed to have dropped off in the years 2019 and 2020 to 5.6 (0.5 Mt and 10.7 Mt) and 3.7 Mt (0.3 Mt and 7.1 Mt), respectively due to periods of lower flows (Figure 3). In 2020, the volume of sand removed, 59 Mt (41 Mt to 75 Mt), was 16 (range of estimate from 7 to 82) times the volume of sand naturally supplied by the basin and riverine transport processes, of 3.7 Mt (0.3 Mt and 7.1 Mt).

The balance between sand extraction and natural replenishment is key to ensuring the sustainability of mining practices and to dampening the negative environmental impacts on the morphology and stability of the river channel (Hackney et al., 2020,

Vasilopoulos et al., 2021). By calculating the natural sand transport of the Mekong River (see section 2.4) we identify temporal variations in this balance at monthly time scales. A monthly time scale is more relevant than a yearly-averaged snapshot because of the seasonality in sediment transport governed by the monsoon that generates distinct periods of increased natural sand supply. Plotting monthly extraction rates alongside rates of sand transport (Figure 5) demonstrates clearly that natural supply is dwarfed by sand extraction, with supply only exceeding removal during two months of the entire five-year period

(August 2018 and September 2019) under the best estimates of extraction. At these times, there is a sand surplus of 1.46 Mt, for both months under this scenario, whilst under the upper bound estimates the supply exceeds removal by 0.6 Mt and 0.8 Mt, respectively. Assuming the lower bound of extraction, sand supply exceeds extraction rate in September 2016 (0.4 Mt), August 2017 (0.2 Mt), August and September 2018 (2.7 Mt combined) and September 2019 (2.1 Mt). Under the upper bound of natural sediment supply, sediment surpluses are still infrequent, occurring within just five months over the five-year period.

**3.2 Spatial variations in sand extraction**

To generate heat maps of mining activity, a kernel density estimate (KDE) was applied to the shapefiles created during the process of identifying active mining vessels. The KDE was calculated with a kernel radius of 1 km and plotted at a resolution of 100 m. Figure 6 shows examples of the heat maps produced for each year between 2016 and 2020 (we provide heat maps of June here as exemplars of the patterns observed; the full collection of monthly maps is provided in the supplementary

information).

A gradual expansion of mining activity is observed throughout the study reach (Figure 6), with patchy, distinct areas of activity seen in June 2016, compared to more continuous activity across the study reach in June 2020. Over this period the area of channel that is actively mined has increased from 54 km$^2$ (15% of the total channel area) to 81 km$^2$ (23% of the total channel

area). Throughout the five year period there are areas of the river that consistently appear as areas of high activity with boat densities of 6 – 8 boats per km$^2$. The two prominent hot spots are at the apex of the Mekong Delta at Phnom Penh, at the confluence-diffluence of the Mekong, Tonle Sap and Bassac Rivers, and around a large island complex at Rokar Korng, located 30 km upstream from Phnom Penh. The latter location saw mining operations begin in 2017 and has since seen consistently high levels of activity. The area around Phnom Penh has been identified by previous research as a hot spot of mining activity

and one which has had significant morphological impact on the river bed as a result (Hackney et al., 2020). This new data reveals that such alterations to the river bed were not a one off event, and the resulting bed incision (estimated at 0.13 m yr$^{-1}$; Hackney et al., 2020) will have been persistent over a period of at least seven years (noting the date of the survey reported in Hackney et al. (2020) was from 2014). Figure 6 also reveals that areas of high activity (both persistent and intermittent) are focussed on areas at the head of islands or at confluences, geomorphic units that are characterised by slower, shallower flows

which promote sediment deposition (Bridge, 1993). Despite being areas of the river channel that may see rapid replenishment of sand due to their morphological characteristics, the excessive rates of extraction, combined with the previously reported sand deficit observed in the Mekong (section 3.1) means that the morphological impact of mining in these locations may result

in changes to local hydrodynamics (Ashraf et al., 2011; Barman et al., 2018) and hence local channel stability. Such alterations to the morphological and hydrodynamical behaviour of the river channel will have attendant consequences for infrastructure stability (for example, loss of housing and road networks, and undermining of bridge piers and foundations), as well as wildlife habitat and agricultural land provision.

## 3.3 Morphological impacts

To highlight the impacts of persistent mining on sections of the Mekong River, a comparison of bathymetric surveys around the island complex at Rokar Korng located 30 km upstream of Phnom Penh (Figure 6) from 2013 and 2019 was undertaken. Over that period bed elevations have incised at a median rate of -0.26 m yr$^{-1}$ whilst mean incision over the same period is -0.16 m yr$^{-1}$ (Figure 7; negative values represent a loss of riverbed sediment due to incision). Incision is predominantly centred on two locations at the upstream and downstream regions of the island complex in the centre of the large meander where incision rates of up to -5 m yr$^{-1}$ are observed (Figure 5a). Prior work has shown that river banks may destabilise if the river bed is incised by >3 m (Hackney et al., 2020). The rates of incision observed here suggest that over the six-year period 2013 – 2019, median (mean) incision is 1.56 m (0.96 m). As such, river banks in this reach are at risk of imminent transition to a period of enhanced erosion resulting from reach scale mining activities. Integrating out the elevation change presented in Figure 7 reveals that 101 Mt of sediment has been lost within this reach over the period 2013 – 2019. This equates to 17 Mt yr$^{-1}$ of sediment being eroded by a combination of natural and anthropogenic activity. Overlaying the locations of mining vessels identified from satellite imagery within this reach over the period 2016 – 2020 reveals that sand extraction is clustered around the locations which have the highest rates of incision (Figure 7b). Thus, it is evident that the major driver of bed incision within this reach is that of sand extraction operating both through the mechanisms of head cut erosion observed by Kondolf (1993) but also through diffusional morphodynamics redistributing sediment around the reach as mining pits infill. It is noted that the period of satellite image analysis (2016 to 2020) is shorter than the period between the bathymetric surveys (2013 to 2019). However, as rates of extraction have been shown to increase over the period 2016 – 2020 (Figures 3 and 5) it is expected that the morphological impacts observed in Figure 7 reflect more recent mining activity and so would be reflected by the vessel activity observed during the period of satellite image analysis.

## 4 Discussion

Providing robust and up-to-date estimates of sediment extraction in riverine and deltaic environments is fundamental to underpin sustainable management of the sand commons. This is especially important in basins such as the Lower Mekong River where the impacts of sand mining are likely to be felt across national borders. The basin-wide impacts of riverbed incision resulting from sand mining are manifested in the LMR as increased river bank instability (Hackney et al., 2020), landward extension of the tidal dominance (Vasilopoulos et al., 2021), saline intrusion (Eslami et al., 2019), and alterations to natural flow regimes (Park et al., 2020; Xin and Park, 2021). Current understanding of the impacts of sand mining on the LMR and its delta are underpinned by the estimates of Bravard et al. (2013). Although the values reported in Bravard et al. (2013)

highlight the negative sand budget the LMR has been experiencing over the past decade or so, findings presented here suggest that they have significantly underestimated the magnitude of this deficit. Indeed, findings reported above (Figure 5) demonstrate that just two months in the past six years have seen positive sand budgets under best estimate scenarios (five months under the lowest estimates).  This not only highlights the need to ensure higher flows are preserved throughout the LMR basin to ensure natural sediment transport is able to replenish sand deposits, but offers potential sustainable management options that focuses on limiting activity at times when sand transport is greatest.

Revised rates of sand extraction are reported here for Cambodia alone. In 2020 these are two times that reported by Bravard et al. (2013) and greater than prior estimates for the entire Mekong basin (50 Mt; Figure 4). It is likely that similar (if not greater) increases in the rates of extraction have been seen in both Laos and Vietnam, and thus we can expect that the current basin-wide extraction volumes are likely to be an order of magnitude greater than those provided by Bravard et al. (2013). For example, Vasilopoulos et al., (2021) quantified a mean channel deepening of 1.6 m between 2008 and 2018 for the principal channels of the Vietnamese Mekong delta, which correspond to a sediment loss of approximately $184\pm39$ Mt a$^{-1}$. It has widely been reported that globally official estimates of sand are limited and frequently underestimated (Bendixen et al., 2019; UNEP, 2019).  Indeed, official records for 2019 reported in Haffner (2020) place extraction rates in Cambodia at 9 million m$^3$ (14.4 Mt). Our analysis suggests that in 2019, ~50 Mt was extracted (35 Mt to 65 Mt), which is 3.5 times (2.4 to 4.5 times) the official reported volumes. Such under-representations of sand extraction in official statistics can have significant implications for local and basin-wide management plans that are underpinned by official records. Existing and future plans for the sustainable management of the Mekong and other large river systems need to be informed by up to date, robust and accurate estimates of sand extraction. The use of high-resolution satellite imagery as detailed here can provide a low-cost, widely accessible, means of providing this underpinning data across a range of environments and at high resolutions and across large spatial and temporal extents.

Figures 6 and 7 highlight the spatial extent of sand mining across the LMR in Cambodia. Hotspots of activity are evident near Phnom Penh and the island complex near Rokar Korng. The former location confirms similar analysis reported in Xin and Park (2021) which identifies areas of high activity at Phnom Penh. Previous work has also highlighted the morphological impact of sand mining in this location revealing large areas of bed disturbance and widespread incision at rates of 0.13 m a$^{-1}$ (Hackney et al., 2020). Similar work throughout the Mekong delta has shown incision rates can be in the order of 0.59 m yr$^{-1}$ over a ten year period (Brunier et al., 2014). This corresponds with rates of incision reported here (0.26 m yr$^{-1}$; Figure 7) which sit between these two previous estimates and strengthens the evidence that sand mining is having major impacts on lowering river bed elevations in the LMR. Importantly, the present research demonstrates that significant riverbed incision is being felt 30 km upstream of the apex of the Mekong delta located at Phnom Penh. The implications of this are significant for the fluxes and the routing of sediment to the apex of the Mekong delta and further downstream where such sediment fluxes are the only counter and offset to eustatic sea-level rise across the delta.  Given that it has already been demonstrated that sand mining is

impacting the hydrology of the Tonle Sap Lake system (Xin and Park, 2021), it is likely that similar impacts are already being

felt along the Mekong and Bassac rivers downstream of the delta apex potentially impacting downstream flood risk and channel stability in the Vietnamese Mekong delta into the future.

**5 Conclusion**

This research provides robust and up-to-date estimates of the volumes of sand removed from the bed of the Mekong River in

Cambodia over the period 2016 – 2020. Using monthly composite images derived from high-resolution PlanetScope satellite imagery, the number and location of mining vessels active on the river are identified. We show that rates of extraction have increased year on year from 24 Mt (17 Mt to 32 Mt) in 2016, to 59 Mt (41 Mt to 75 Mt) in 2020 at a rate of ~8 Mt $yr^{-1}$ (6 Mt $yr^{-1}$ to 10 Mt $yr^{-1}$), where values in parenthesis relate to lower and upper error bounds respectively. Our revised extraction estimates for 2020 (59 Mt) are two times greater than previous best estimates for sand extraction for Cambodia (32 Mt) and

greater than current best estimates for the entire Mekong Basin (50 Mt). We show that areas of high vessel activity are correlated to areas of major bed incision resulting from mining activities and demonstrate median bed incision of -0.26 m $yr^{-1}$ over the period 2013 to 2016. Our revised estimates highlight the need for regular monitoring of sand mining activity and advocate for local and basin management plans to be updated to better reflect the current situation. The tools developed herein provide a low-cost, robust method to provide regular up-to-date estimates of the volume and location of mining activity in the

world's rivers and deltas, and offer a step-change in the insight into the environmental and biophysical implication of the sand extraction industry.

**Data availability**

All shapefiles produced from PlanetScope imagery are available on request from corresponding author. Hydrological data for the gauging stations in Cambodia are available from the Mekong River Commission Data Portal

(http://portal.mrcmekong.org/).

**Author contribution**

**Christopher Hackney:** Conceptualisation, Methodology, Investigation, Formal analysis, Writing – Original Draft, Supervision. **Grigorios Vasilopoulos:** Conceptualisation, Methodology, Investigation, Writing – Review and Editing, Supervision. **Sokchhay Heng:** Methodology, Investigation, Writing – Review and Editing. **Vasudha Darbari:** Methodology,

Investigation, Writing – Review and Editing. **Samuel Walker:** Methodology, Investigation, Writing – Review and Editing. **Daniel R. Parsons:** Conceptualisation, Methodology, Investigation, Writing – Review and Editing.

**Competing interests**

The authors declare they have no conflict of interest

**Acknowledgements**

CRH was supported by a NUAcT Fellowship from Newcastle University and the UKRI GCRF Living Deltas Hub (grant NE/S008926/1). CRH and DP acknowledge funding from NERC (NE/JO21970/1 and NE/JO21881/1; to Southampton and Hull, respectively). GV, VD, and SW, were supported by the University of Hull's GCRF funding and an Energy and Environment Institute scholarship. DP acknowledges funding from EU Horizon 2020 Programme (ERC GEOSTICK, 725955).

We extend our thanks to the Department of Hydrology and River Works, Cambodia for their help and field assistance in collecting the bathymetry data in 2013. We thank the Mekong River Commission for their help in accessing and providing water level and discharge data used in this study.

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

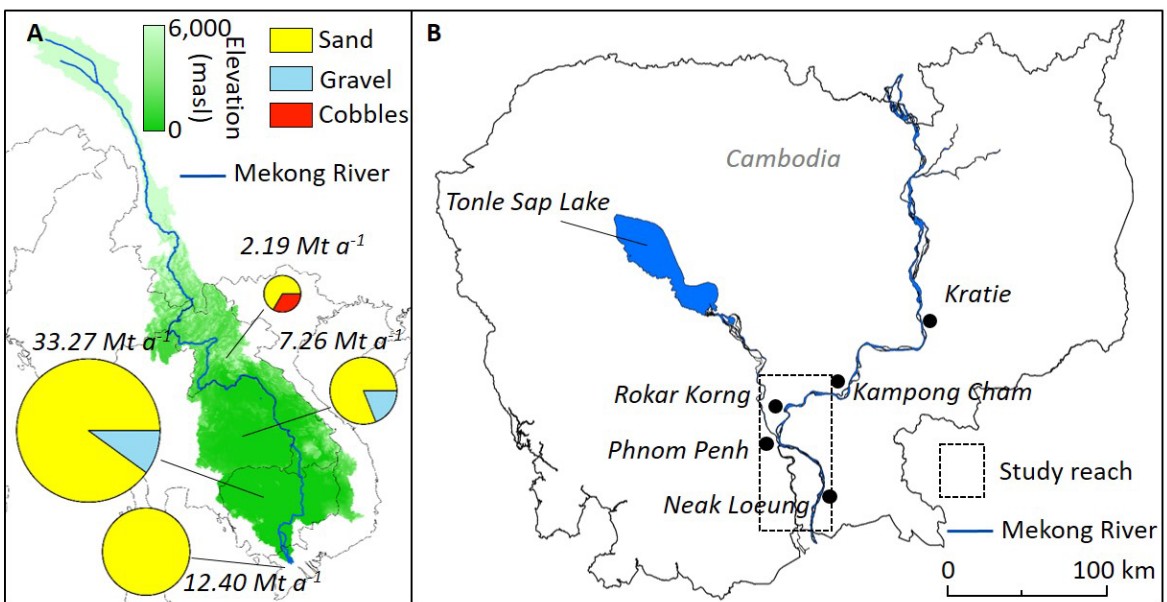

**Figure 1: A) Established sediment extraction rates (Mt yr⁻¹) across the Mekong River basin in 2013. The composition and volume of material extracted are disaggregated by country taken from Bravard et al. (2013). The size of the pie charts scales with the volume extracted. B) Location of the study reach in Cambodia (dashed box) from Kampong Cham to the Vietnamese border. Major river**
**channels and lakes are highlighted by blue lines.**

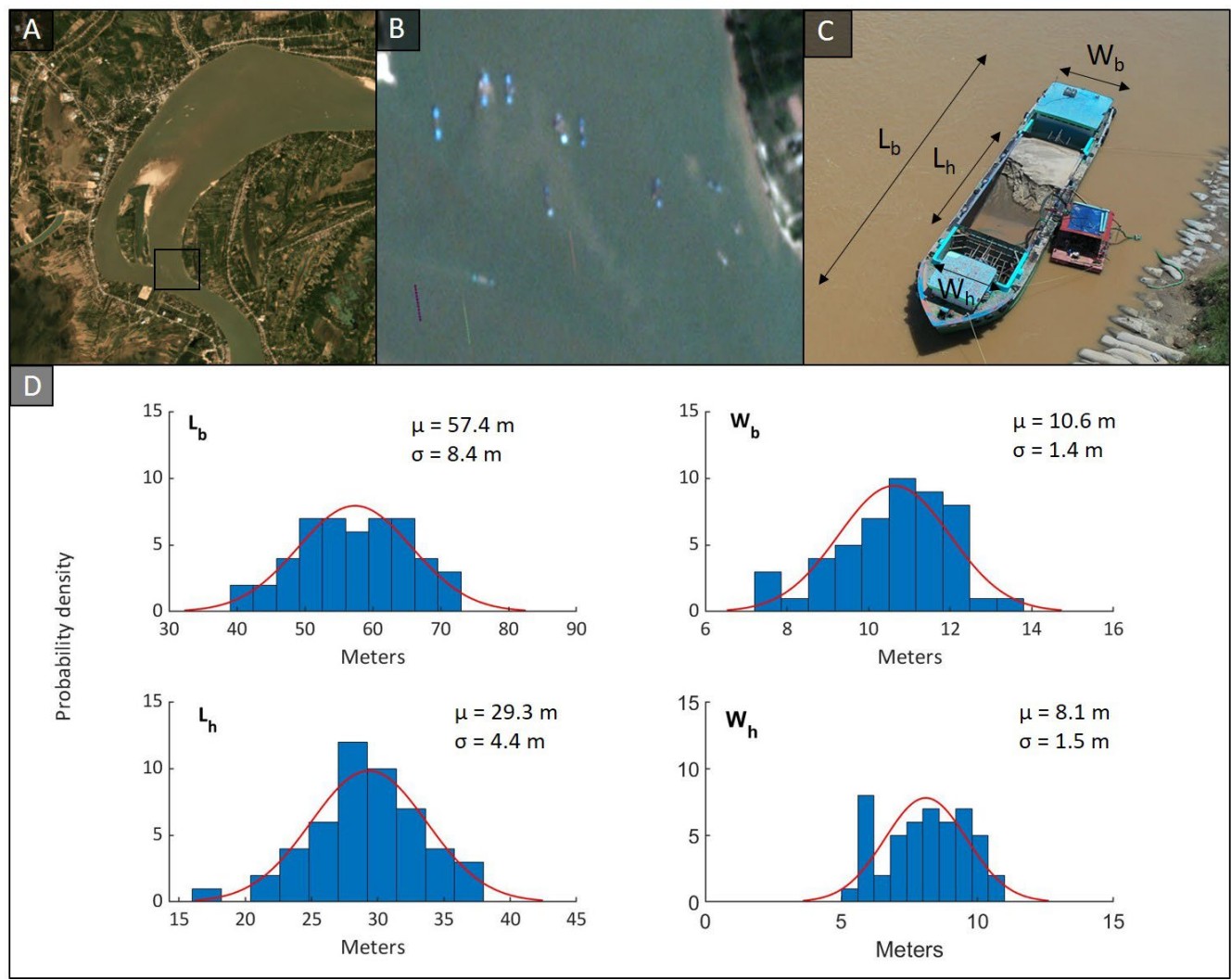

**Figure 2: A) Example PlanetScope monthly composite image of a stretch of the Mekong river from December 2018 (Planet Team, 2018). The square highlights the area depicted in panel B. B) Zoomed in area of PlanetScope imagery (Planet Team, 2018) for the Mekong showing how mining vessels are identifiable within the satellite imagery. C) Dimensions of mining vessels recorded including the boat length ($L_b$), the boat width ($W_b$), the hold length ($L_h$) and the hold width ($W_h$). These latter two dimensions are important for the calculation of extraction volumes as they define the volume of sand each boat can transport when fully laden. Image is the authors own. D) Estimated dimensions of the mining vessels active on the Mekong taken from Google Earth imagery © Google Earth. 50 boats were measured and dimensions were recorded as depicted in panel C (µ = mean, σ = standard deviation).**



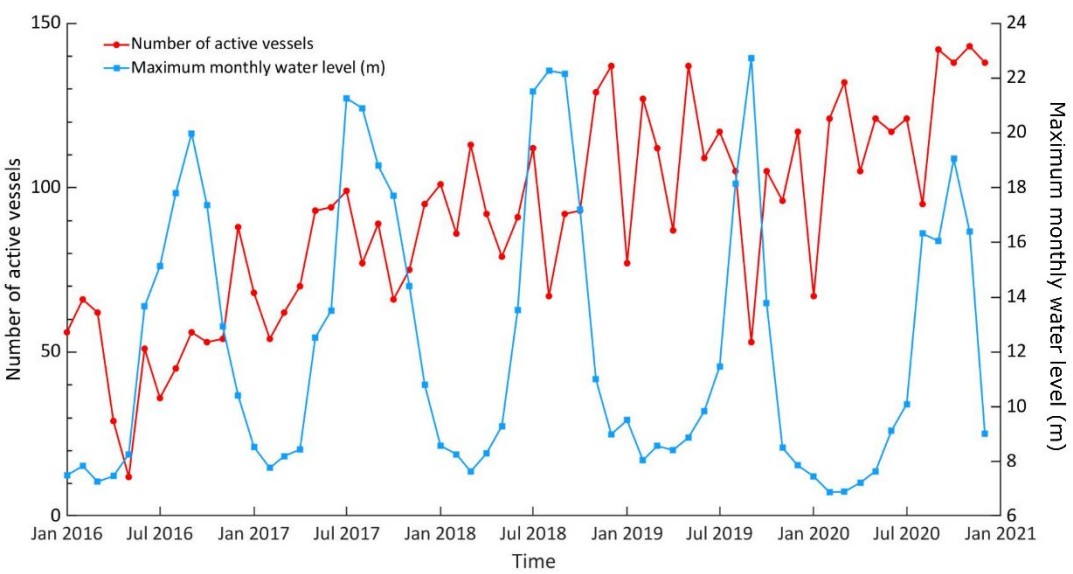

**Figure 3: The number of active vessels (vessels greater than 100 m from the riverbank) identified in each monthly composite image over the period January 2016 to December 2020 (red) and the maximum water level (m) at Kratie in each month over the same period, highlighting that higher water levels are associated with reduced river vessel traffic due to difficult navigation conditions and faster flowing water in monsoon periods.**


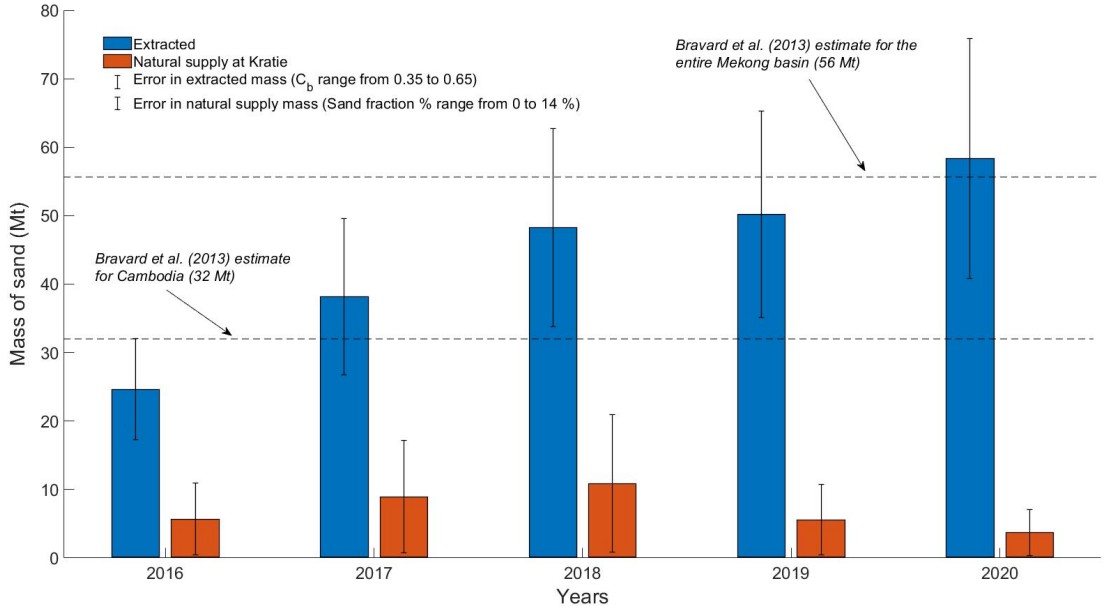

**Figure 4: Volume of sand extracted each year for the period 2016 to 2020 (blue bars) for the study reach in Cambodia between Kampong Cham and the Vietnamese border. Error bars represent annual volumes calculated under varying levels of the compaction factor, $C_b$, from 0.35 to 0.65 (see section 2.3 for details). Orange bars represent the annual volume of sand supply transported naturally by riverine processes at Kratie, Cambodia. Error bars represent annual volumes calculated under varying levels of the proportion of suspended sediment flux that is sand (ranging from 0 to 14 %). The mass of sand extracted from Cambodia in 2013 as estimated by Bravard et al. (2013) is demarked by the dash-dot line at 32 Mt, similarly the estimated mass of sand extracted from the entire Mekong basin in 2013 is demarked by the dashed line (56 Mt).**



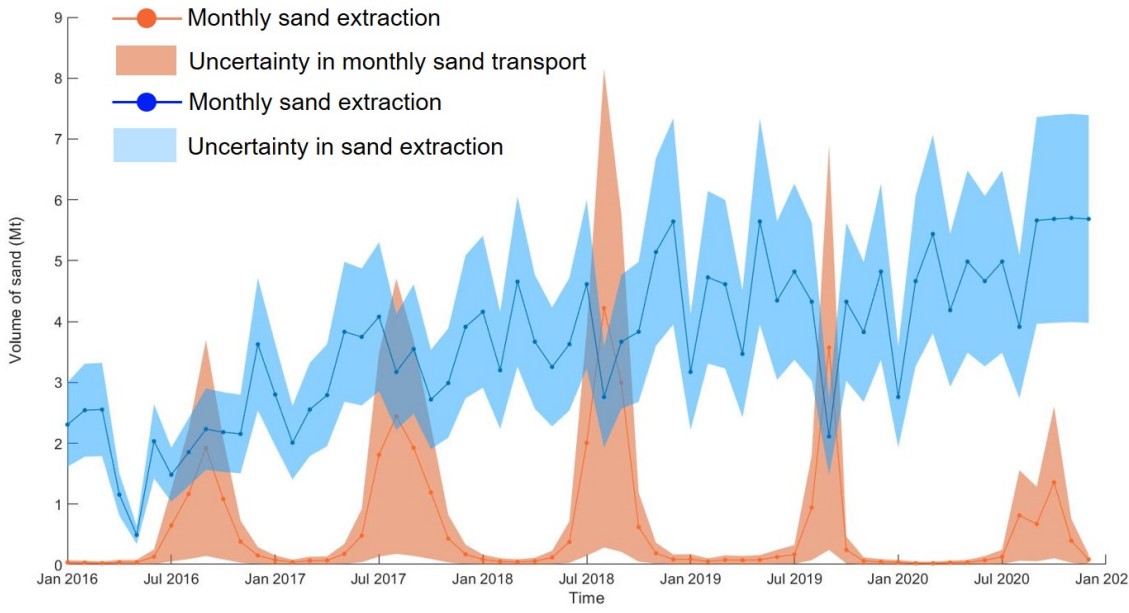

**Figure 5: Monthly volumes of sand extraction (Mt) estimated from the number of vessels visible in the satellite imagery over the period 2016 to 2020 (blue) with error estimates defined by adjusting the compaction coefficient, $C_b$, between 0.35 and 0.65 (blue shaded area). Monthly natural sand transport (Mt) at Kratie (see section 2.4 for details) for the same period (orange) with error estimates derived by adjusting the percentage of suspended sand fraction from 0 to 14% (around a mean of 7% - orange area).**


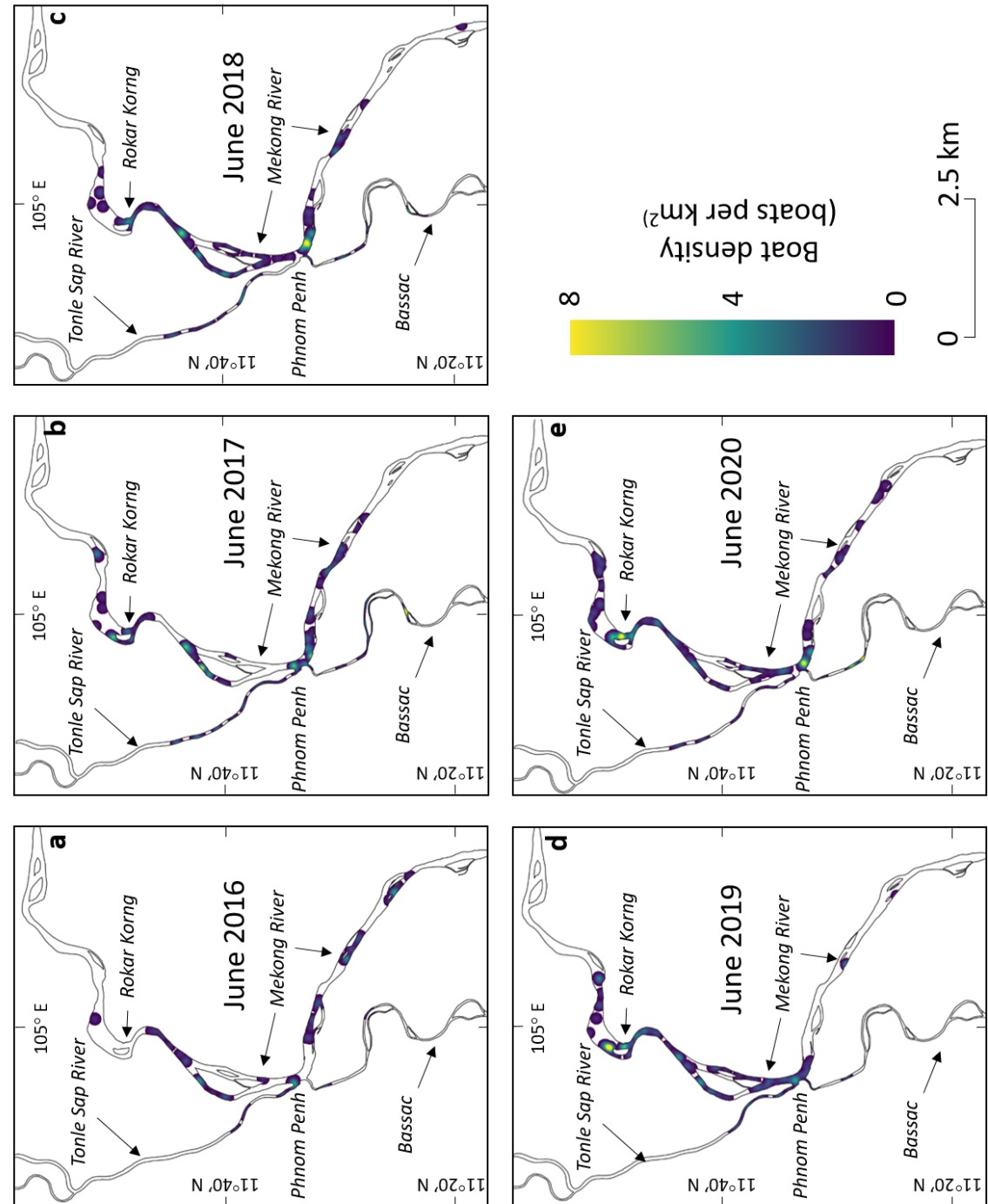

**Figure 6: Annual heat maps for June for the period 2016 - 2020 of the Mekong River around Phnom Penh showing the locations of mining activity and the density of boats (boats per km$^2$) showing changes in the spatial distribution of mining activity over the five year period.**

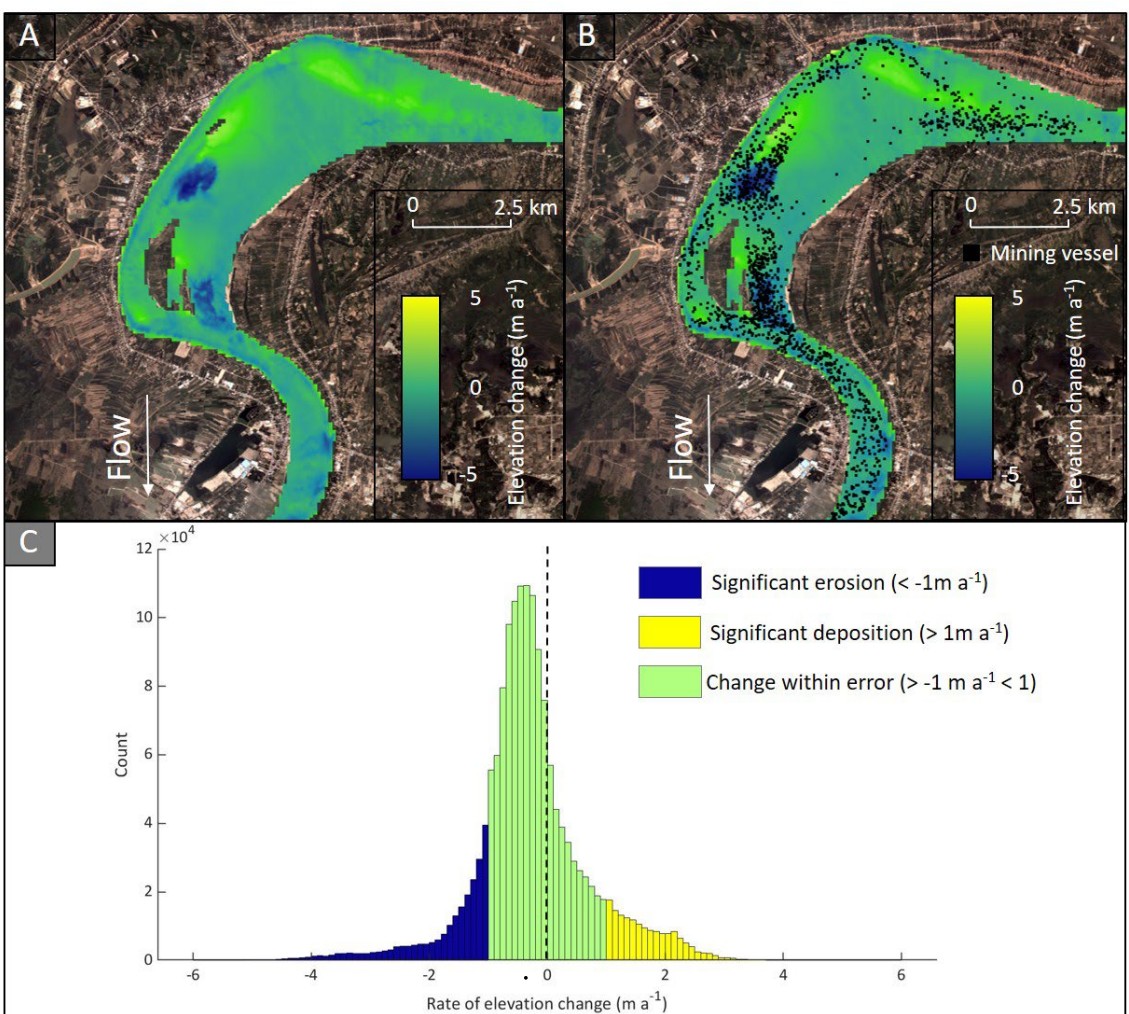

Figure 7: A) Rate of elevation change between 2013 and 2019 of the Mekong River near Rokar Korng highlighting areas of erosion (blue) and deposition (yellow). B) Overlain on the map depicted in A are the locations of all observed mining vessels during the period 2016 to 2020 demonstrating that the zones of greatest erosion observed in A correspond to areas of high mining activity. C) Histogram of elevation changes for this reach between 2013 and 2019. The median rate of bed elevation change is -0.25 m yr⁻¹ (mean elevation change is -0.16 m yr⁻¹). The dashed line demarks zero net change in elevation. Image background in panels A and B are PlanetScope monthly composite scenes from December 2020 (Planet Team, 2018).