# Peer review of "Sand mining far outpaces natural supply in a large alluvial river"

_Earth Surface Dynamics, 2021_

## Author Comment (AC1)

**Review 1:**

The paper discusses the extremely pertinent topic of over-harvesting of rivers through sand mining. The region of focus chosen by the research team is particularly important as it is one of the highest impact global regions with respect to sand mining. Given the relatively limited academic literature around quantification of sand mining from different river ecosystems across the world, this paper makes an extremely important contribution to both literature and data on sand mining. I was very excited to read this paper and the figures that emerge from it. This paper provides new, and higher numbers which are highly likely to much closer to the reality of sand extraction in the LMK than previous estimates. I also hope to see further work from the authors as this is important, urgent and necessary data.

**We thank the reviewer for the time spent providing the review and for the positive comments regarding the manuscript.**

However, I am not an expert in GIS mapping and the other technical aspects of this research paper, and hence cannot comment on the details of the scientific assessment process. I hope other relevant reviewers can address technical issues and preform the quantitative check.

The abstract needs some grammatical improvements.

P1, L10: ..in part due to (followed by only one reason)... seems incomplete //either more "reasons" could be added here or there could be a minor edit to this statement.

We disagree. The use of 'in part due' followed by a single reason is common and is used here to highlight the fact that the nature of the extractive process is one of the main reasons why such estimates are lacking.

Next line: grammatically incorrect sentence: Can it be replaced with...?: Current estimates, based on 2013 figures, indicate that basin wide sand extraction in the Mekong River stood at least 50 Mt.

Agreed, we have changed this sentence to the following "For the Mekong river, the widely assumed estimate of basin-wide sand extraction is 50 Mt a year. This figure is based on 2013 estimates and is likely to be outdated."

"Year on year" could be changed to: on a yearly basis? Or just "yearly"?

Agreed, we have removed "year on year" and replaced with "yearly"

"We use.....respectively" is a very long statement and can easily be broken down into two or even three to enhance clarity for the reader.

Agreed, this was a long and confusing sentence. We have broken it down into two sentences as follows "We use monthly composite images from PlanetScope imagery (5 m resolution) to estimate sand extraction volumes over the period 2016 - 2020. We show that rates of extraction have increased on a yearly basis from 24 Mt (17 Mt to 32) in 2016, to 59 Mt (41 Mt to 75 Mt) in 2020 at a rate of ~8 Mt yr-1 (6 Mt yr-1 to 10 Mt yr-1); where values in parenthesis relate to lower and upper error bounds, respectively."

P1, L29: Although it is correct that massive growth has been observed in the last three decades (around 1990s with neoliberal reforms being pushed through post Washington Consensus, trends

show the increase since 1970s already). It might be interesting to include this paper... See Miatto et. al., 2016 paper quantifying global non-metallic mineral consumption.

We thank the reviewer for spotting this, and agree that aggregate trends have been on the rise longer than suggested in the text. We have amended the text to reflect this ("...unprecedented growth since the 1970s") and have added the suggested reference here too.

P1, L30: Although correct, it might be better to specify: Estimates indicate that currently/today, at minimum, between 32-50 billion tons of aggregates...

Agreed, we have amended the text to include this clarification, and have added reference to the recommended paper by Bisht and also the work by Koehneken et al. (2020) here too.

P2, L32: ...; with impacts manifesting as... statement could end with ..., among others. Given the many more serious ecological and environmental impacts of the over-extraction of sand. You could see/quote Bisht, 2021 paper on global sand extractivism for this.

We have added the clarifier "amongst others" to the end of the sentence as recommended.

**P2, L35: is reaches correct? Or is it "regions"?**

We have altered this to "regions" as on reflection "reaches" was being used in a fluvial context which may not have been appropriate for the broader narrative of this paragraph. We feel regions better captures this.

P2, L39: ...pervasive and diffused...

We have left this as "diffuse" as it is used here an adjective rather than in the past participle.

P2, L56: "changed"

**Agreed, amended.**

P2, L54-56. The statement is too long and seems a little unclear to me (The last part is referring to the recent bans in extraction and export of sand to Singapore?). Can this be split at "...(Hackney, 2020). However, tighter contractions on locations and rates of extraction, ...., are currently lacking...

Agreed, we have split the sentence in two at the reviewer's recommendation.

P2, L62: could "sand transport" here be replaced with another phrase? Maybe, sediment transport. I was a bit confused about whether this meant the amount of sand being transported out of the LMK, until I read the details of the methodology below.

We have left this as it is as we mean to refer explicitly to the transport of sand (as is later described in the methods). Indeed, later on in the sentence (LN 63) we do mention that we are looking at "the volume of sand naturally transported by the river". Changing this to "sediment transport" would not be reflective of what we have done and would add more confusion by including other grain sizes in to the mix.

Review 2:

This paper quantifies sand extraction from the lower Mekong River using remote sensing. The authors document the number and dimensions of sand mining vessels from PlanetScope and Google Earth images respectively. Based on sand mining vessel dimension and number, the authors estimate the volume of sand extracted each month from 2016–2020. The extraction rate is compared to the total sand transport estimated from previous studies, and it is determined that the extraction far exceeds transport. As a consequence, the LMR bed is scouring. The authors demonstrate that the deepest scours of the channel bed correspond to the most active areas of mining, and therefore conclude that the primary cause of these bathymetric lows is driven by sand mining. This paper offers a simple method to estimate sand mining in the world's large rivers. The content is timely, as the industrial demand for sand continuously increases.

**Comments:**

The authors nicely demonstrate a correlation between intense sand mining and localized channel bed scour (e.g., Figure 7). It would be helpful if the authors made an explicit connection between these highly localized observations and the broader impacts of sand mining outlined in the introduction (L48–51). Do the scour locations here correspond to enhanced bank erosion or other morphological responses?

In lines 48 – 51 we refer to a range of broader impacts of sand mining, including salinity intrusion and extent, changing flood frequencies, regional fish stocks and alterations to the hydrology of the Tonle Sap Lake. To assess the contribution of incision to these processes would require a much broader spatial and temporal assessment than this study can offer. For example, salinity intrusion is limited to the delta region, approx. 250 km downstream of our study site, whilst the impacts on the Tonle Sap flood regime would require studies of bed morphological change around Phnom Penh along with longer term assessments of hydrology within the lake system. It is noted here that other studies (Eslami et al., 2020; Vasilopoulos et al., 2021; Xi and Park, 2021) have addressed these relationships and we refer to these within the manuscript. Providing an explicit link to enhanced bank erosion rates is also beyond the scope of this study and detailed hydrodynamic and bank geotechnical data would be necessary to disentangle any enhanced erosion resulting from bed incision from rates of natural erosion. We note that in an earlier study (Hackney et al., 2020) the link between bed incision and enhanced bank erosion rates in the Cambodian Mekong River was made, with the authors highlighting that an incision of 3 meters is likely to induce reach wide instability of the river banks. To link this work through to our results on incision rates here, we have added an additional section to the morphological impacts section (lines 242 - 245) in the revised manuscript) that places the rates of incision observed here in the context of this prior work on bank instability.

It would be helpful for the authors to explicitly state why Planet data are used to estimate the number of mining vessels and aerial images from Google Earth are used to get dimension. Why not use the daily images on Planet to estimate dimension? Presumably this is due to resolution, but the resolution of the Google Earth images is not stated.

We thank the reviewer for raising this point and we have added a point of clarification to this effect in section 2.3 (In 113 in the revised manuscript) that reads "Google Earth imagery was used to constrain the vessel dimension as its higher spatial-resolution permits greater confidence in providing robust dimensions compared to the Planetscope imagery".

Are the scours filled at all during low mining times?

This is an interesting and insightful comment, and one that is really important to understand with regards to the sustainability of the sand resource. We think that when sand mining at a specific

location stops (for example when a miner is relocated to a different area), scours get gradually filled by a combination of downstream sediment transport (although previous work (Hackney et al., 2020) has shown this to be limited in the case of the Mekong) and the reworking of sediment from areas surrounding the scours. This leads to a net incision of the wider river reach which we discuss in Lines 237-252 (and also highlighted in Hackney et al., 2020). Although the data presented here are not sufficient to test this hypothesis for our study area (given the lack of repeat surveys over small – weeks to months – time frames), a similar mechanism of infilling of sand mining pits by eroding sediments from the surrounding channel beds has been described by Jordan et al., (https://doi.org/10.1038/s41598-019-53804-z) in Vietnam. We direct the reviewer to line 247 of the revised manuscript which links the infilling of pits to observed incision rates.

Line Edits:

L12: remove "and". "...sand extraction are 50 Mt based on estimates from 2013."

This section has been reworded in response to similar comments from Reviewer 1

L21: specify if this incision is localized or widespread.

We have added the clarification that the incision is "reach wide" here as given this is the abstract we are limited for words. However, we feel that the use of "reach wide" shows that although localised to the reach of study, the incision is being felt across this reach of river and so may be considered widespread.

L108: Please provide more information about the aerial images. How many images were used? What dates do the images span? Resolution?

We thank the reviewer for point this out. We have added details relating to the Google Earth images to the text at this point. Specifically, we use imagery taken on 24th September 2019 available in Google Earth as this image covered the entire area of study ensuring that the vessels collected were measured within one day. Although the specific image resolution cannot be determined. The image provide CNES/Airbus routinely provides images of resolutions up to 15 cm to Google Earth.

L120: Is it assumed that mining vessels are active 7 days a week?

This is correct, and we have clarified this in the text.

L127: Be consistent when reporting ranges (e.g., L127 "2016 - 2020" vs. L59 "2016-2020")

We thank the reviewer for pointing this out but note that there is a consistent one space between all ranges throughout the text. The appearance of additional space is due to the requirement to fully justify text in the paragraph on line 127 which has forced greater spacing between words.

155: Be consistent reporting negative values. Sometimes a space is included, sometimes not (e.g., L157 "- 0.6 m" vs. L21 "-0.26 m"

We thank the reviewer for pointing this out and have removed the additional gap after the minus sign on line 157.

**L193: "negate the negative" – Improve word choice**

We agree with the reviewer that the choice of the word negate was problematic. We have amended this to "dampen" in acknowledgment that mining impacts are likely to be felt regardless, but that with a replenishment of sand, these impacts may be reduced.

L187–188: Please clarify this sentence. What is meant by "basin sand supply"? Sand supply to the basin?

We agree the choice of "basin" was confusing. We have amended this to read "natural sand supply".

L216: Should read "saw mining operations begin in 2017" rather than "saw mining operations began in 2017"

**Agreed and amended.**

L228: "Attendant"? Word choice

We feel the use of the word here is correct as it has the meaning "occurring with or as a result of".

283: Label the apex on Figure 1

We have added clarification in the text that the apex of the delta (the first bifurcation) is located at Phnom Penh. Phnom Penh is labelled on figure 1b.

Figure 1 caption: It's not clear what the author means by "basin country."

The word basin has been removed in this caption.

Figure 3 caption: State why water level is important here. (High water levels correspond to low vessel traffic).

Agreed, the following text has been added to the caption of figure 3... "highlighting that higher water levels are associated with reduced river vessel traffic due to difficult navigation conditions and faster flowing water in monsoon periods."

Figure 4: Please state the source of data for the natural sediment supply at Kratie. Also, please choose a color-blind friendly palette.

We thank the reviewer for raising the issue of colour blindness. We have recoloured the bars in Figure 4 to be coloured by a colour pairing that maintains their contrast. This has also allowed us to recolour so that the bar colours for natural sand supply in figure 4 and 5 are now consistent. We have also added to the legend that the natural sand supply is at Kratie as recommended by the reviewer.

Figure 7 L453: The word "change" is missing. The median rate of bed elevation...(insert change)

Agreed, the word "change" has been added.

---

## Author Response (AR2)

Dear Editor,

Thank you for the thorough and constructive comments on the manuscript. We have addressed each of the points raised and amended the text accordingly and feel the manuscript is much improved as a result. Below is a point-by-point response to the points raised, with the original comments in roman black text, and our responses in italicised red text. In our response we have highlighted the line numbers in the revised manuscript at which point we have made edits to the text. We also note here that we have added the © Google Earth copyright icon as requested by the file validity check, to figure 2.

Yours faithfully,

Dr Chris Hackney (on behalf of all authors)

Line 100 – Please explain why a 100 m buffer? What is the typical channel width – useful to specify here in the context of that buffer

*We agree the rationale behind the 100 m buffer was not clear and the text (Line 106 – 111) has been amended to better reflect the choices behind this decision. The reason a buffer is applied is to remove vessels from the analysis which may be moored on the bank and therefore not active in any mining operation on the day the satellite image was capture. The mining vessels have a mean length of 60 m (see line 95 in the revised manuscript and also figure 2), although the longest vessels my reach up to 75 m in length. The 100 m buffer was chosen to ensure that if moored perpendicular to the bank, that the vessel would be fully contained within the buffer. From visual observations it has also been noted that active mining operations do not occur along the near bank zone, and so it is unlikely that a buffer of 100 m would capture active operations. Along the reach of the river in the study area, the channel width varies between 600 m at its narrowest to 2,500 m at its widest – although typically channel widths are around 900 m. A 100 m buffer on each bank, therefore accounts for between 8 and 33% (though typically 22%) of the channel width ensuring the majority of the channel, and the areas that are actively mined remain accounted for.*

Line 111 – The density seems low for quartz sand, or does this value already account for porosity?

*The density value (1,600 kg m$^3$) used here was taken from the previous study of sand extraction from the Mekong (Bravard et al., 2013) to ensure comparability between the estimates arrived at. This values is for dry sand. We have amended the text at line 128 in the revised manuscript to make it clear that we have used the same density value for comparison with Bravard et al.'s estimates and also commented that varying the density to that of quartz sand (2,650 kg m3) would result in a 65% increase in tonnage estimates.*

Line 133 – How variable is this sand fraction likely to be?
*We note that this relates to suspended sand fraction which is perhaps less important with respect to the morphological impacts when compared to bedload sand transport, but is still important for overall riverine sustainability and in calculating the sand deficit. Prior fieldwork across a range of discharges (14,500 to 55,000 m$^3$ s$^{-1}$) and a range of study sites located (reported in Hackney et al., 2020, Nature Sustainability) within the study region shows that the range of sand fraction within the water column does range from 1 to 14% (averaging out across the study area at 7%). Thus it is likely that the sand fraction varies spatially and temporally (as discharge varies). We acknowledge that this variation will lead to substantial changes in the estimates of the deficit (as mentioned in the comment below) and that locally, sand deficits may be greater and lower depending on the availability of sand in suspension. We have added comments on this to the revised manuscript (Line 151 in the revised manuscript). However, we also note that in terms of a reach scale assessment of the impacts of sand mining on river morphodynamics, it is necessary to report an average value as it is difficult to assess the spatial variations in deficit without more spatially explicit estimations of bed load transport.*

Line 135 – Say something about how good this empirical bedload transport model performs based on the previous work?

*We agree with the editor that more details as to the performance of the bedload transport function used here are needed. We direct the reader to Hackney et al. (2020) where the model is first presented for full details but note in the revised manuscript (lines 164 - 168) that the empirical bedload function used is based on that of Bagnold (1980). Statistically, the model has an r2 consistent with that of Bagnold's (1980) model and other empirical bedload transport functions widely used ($r^2$ = 0.22, P < 0.1) based on 12 observations across a range of discharges covering 78% of all bedload transport events during the period 1980 - 2015. As such, given the ever difficult nature of characterising bedload sediment transport, we have confidence that the model used here is appropriate for the study reach. Given the small fraction of the total sand load that is bedload (< 1%, see Hackney et al., 2020) the uncertainty in bedload transport estimates calculated using the 95% confidence intervals around the transport predictor fall within the adjusted values of sand content resulting from varying the sand fraction transported in suspension (see comments above and below). Thus, the error associated here is contained within the new error bars presented in Figure 4 and 5 (see comment below). We have added text in the manuscript detailing this, and also commenting that it is the sand fraction in suspension that is the biggest source of uncertainty in estimating the influx of sand through natural transport processes (line 169 - 171 in the revised manuscript).*

Line 159 – What is the M9 instrument? Its acoustic emission frequency?

*As stated in Line 176 of the revised manuscript, the M9 instrument is an acoustic Doppler current profiler. We have specified the manufacturer (Sontek) in line 159 to better reflect that this is a brand and allow the reader to search for the instrument if they wish. The bathymetry was collected using the 1 MHz vertical beam on the instrument and this detail has been specified in the text of the revised manuscript (line 179).*

Line 185 – How likely is it that he 7% sand has varied, this has some leverage in these calculations. For instance, could it be double or half that value (i.e. 10.9 Mt sand becomes 21 Mt at 14%). Or put another way, its clear the deficit is there, but what is the major uncertainty source on the sediment flux estimate, and is it likely to be conservative or not?

*We agree that this value is important to accurately quantify and constrain the sand deficit. As mentioned in the response to the editor's prior comment, the suspended sand fraction does vary in space and time, from 0 to 14 % (averaging out at 7%). To reflect this and highlight the importance of this variation in the estimation of natural sand transport, we have added error bars on to Figure 4 and added shaded error areas to Figure 5 (and altered the text reporting these results) to reflect this uncertainty and highlight the potential natural sand transport that would occur if the estimate of suspended sand varied from 0 – 14% (as our prior work suggests is possible). As noted in the comment above re bedload transport estimates, it is this fraction which provides the greatest uncertainty in estimating natural sediment loads. This point is further highlighted in the text (see comment above for details).*

---

## Author Response (AR3)

Dear Editor,

Thank you for your thorough and constructive comments on the manuscript. We have addressed each of the points raised and amended the text accordingly and feel the manuscript is much improved as a result. We have implemented all the minor grammatical corrections suggested by the editor. Below is a point-by-point response to the more substantive points raised, with the original comments in roman black text, and our responses in italicised red text. In our response we have highlighted the line numbers in the revised manuscript at which point we have made edits to the text. We also note here that we have added the © Google Earth copyright icon as requested by the file validity check, to the figure 2 caption, but note that none of the figure panels display Google Earth Imagery, rather panels A and B display PlanetScope imagery (which is credited in the caption) whilst panel C is an aerial image taken by the lead author.

Yours faithfully,

Dr Chris Hackney (on behalf of all authors)

Point-by-point response:

Ln 54: I wonder if it would help at the outset to specify the density you assume in the volume-to-mass conversions throughout.

*We thank the editor for this comment and agree it would be useful to specify this detail here. We have altered Ln 54 in the revised manuscript so that it now reads "31 million m3 yr-1 assuming a density of sand of 1,600 kg m-3 55 following Bravard et al., 2013".*

Ln 63: Addition of "we" needed

*Agreed and corrected.*

Ln 85: Delete "will increase"

*Agreed and deleted.*

Ln 100: Addition of "related too"

*Agreed and added.*

Ln 102: Addition of "is"

*Agreed and added.*

Ln 133: Suggest deletion of "sensitive and"

*Agreed and deleted.*

Ln 160:  Delete "at"

*Agreed and deleted.*

Ln 164: Addition of "the method"

*Agreed and added.*

Ln 189: Addition of "(ADCP)"

*Agreed and added.*

Ln 210: Addition of "increase of"

*Agreed and added.*

Ln 210: Change "This value" to "The 2020 extraction volume"

*Agreed and changed.*

Ln 220: Addition of "range of estimates of"

*Agreed and added.*

Ln 210: Change "undergone" to "had"

*Agreed and changed.*

Ln 267: Addition of ": negative values represent a loss of riverbed sediment due to incision" to clarify what that negative means incision not deposition"

*We agree this change is needed to avoid potential confusion with the reader and have made the change accordingly.*

Ln 332: Addition of "Mt"

*Agreed and changed.*

Figure 1 caption: Presumably other black lines are rivers?

*We thank the editor for highlighting this and have altered the caption in the revised manuscript to say "Major river channels and lakes are highlighted by blue lines" to avoid any confusion.*

Figure 2 caption: Addition of "μ = mean, σ = standard deviation)"

*We thank the editor for suggesting this and have added this clarification to the end of the figure caption.*

Figure 5: Presumably the legend should read "sand transport"

*We thank the reviewer for noticing this. Yes, there was a typo in the figure legend, this has now been corrected to correctly report that the orange shaded area represents sand transport.*

Figure 6: Maybe add another longitude mark in each panel for scale

*Unfortunately, the scaling and location of the map within the panels means that an addition longitude mark would sit almost exactly on the right hand side box boundary of each panel. Having tried to add an additional mark, we have decided that the outcome looks messy and prefer the original figures. Therefore we have not made any changes to figure 6.*